# Direct visualization of a Fe(IV)–OH intermediate in a heme enzyme

Hanna Kwon[1], Jaswir Basran[1], Cecilia M. Casadei[1,2], Alistair J. Fielding[3], Tobias E. Schrader[4], Andreas Ostermann[5], Juliette M. Devos[2], Pierre Aller[6], Matthew P. Blakeley[2], Peter C.E. Moody[1] & Emma L. Raven[7]

Catalytic heme enzymes carry out a wide range of oxidations in biology. They have in common a mechanism that requires formation of highly oxidized ferryl intermediates. It is these ferryl intermediates that provide the catalytic engine to drive the biological activity. Unravelling the nature of the ferryl species is of fundamental and widespread importance. The essential question is whether the ferryl is best described as a Fe(IV)=O or a Fe(IV)–OH species, but previous spectroscopic and X-ray crystallographic studies have not been able to unambiguously differentiate between the two species. Here we use a different approach. We report a neutron crystal structure of the ferryl intermediate in Compound II of a heme peroxidase; the structure allows the protonation states of the ferryl heme to be directly observed. This, together with pre-steady state kinetic analyses, electron paramagnetic resonance spectroscopy and single crystal X-ray fluorescence, identifies a Fe(IV)–OH species as the reactive intermediate. The structure establishes a precedent for the formation of Fe(IV)–OH in a peroxidase.

[1] Department of Molecular and Cell Biology and Leicester Institute of Structural and Chemical Biology, University of Leicester, Lancaster Road, Leicester LE1 9HN, UK. [2] Institut Laue-Langevin, 71 Avenue des Martyrs, 38000 Grenoble, France. [3] The Photon Science Institute and School of Chemistry, The University of Manchester, Manchester M13 9PL, UK. [4] Jülich Centre for Neutron Science (JCNS) at Heinz Maier-Leibnitz Zentrum (MLZ), Forschungszentrum Jülich GmbH, Lichtenbergstr. 1, 85748 Garching, Germany. [5] Heinz Maier-Leibnitz Zentrum (MLZ), Technische Universität München, Lichtenbergstr. 1, D-85748 Garching, Germany. [6] Diamond Light Source Ltd, Diamond House, Harwell Science and Innovation Campus, Didcot, Oxfordshire OX11 0DE, UK. [7] Department of Chemistry and Leicester Institute of Structural and Chemical Biology, University of Leicester, University Road, Leicester LE1 9HN, UK. Correspondence and requests for materials should be addressed to P.C.E.M. (email: peter.moody@le.ac.uk) or to E.L.R. (email: emma.raven@le.ac.uk).

The family of heme-containing peroxidase enzymes is widespread in biology. They catalyse the $H_2O_2$-dependent oxidation of a range of different substrates, and in doing so underpin a number of essential biological processes in bacterial, yeast, plant, fungal and mammalian systems[1,2]. The key to their catalytic power is the formation of two transient, oxidized heme intermediates. These intermediates—which are both $Fe^{IV}$ (ferryl) species but differ in the oxidation state of the porphyrin ring—form sequentially during catalysis. Both intermediates were originally observed in horseradish peroxidase but were mistakenly interpreted as enzyme–substrate complexes: the first (green) intermediate was discovered by Theorell, and the second (red) intermediate by Keilin and Mann[3,4]. They were eventually given the names Compound I and Compound II to differentiate them from the enzyme–substrate complex[5,6]. These same two intermediates are used widely in numerous other $O_2$-dependent catalytic heme enzymes, most notably the cytochrome P450s, the nitric oxide synthases, the terminal oxidases plus the heme dioxygenases.

Many years have passed since the first observations, but establishing the nature of these transient ferryl species remains a fundamental question[7–10]. A particular focus has been the nature of the ferryl group in Compounds I and II, and whether it is best described as an Fe(IV)=O or a Fe(IV)–OH species. This is important because the bonding interactions and the protonation state of the ligand bound to the iron controls the reactivity—and hence the biological usefulness—of each intermediate. But this conceptually simple question has proved fiendishly difficult to answer, and thus has become highly controversial. Part of the problem is that spectroscopic approaches—which have mainly used EXAFS, resonance Raman, and Mossbauer—are only indirect reporters of the protonation state of the ligand; none can directly visualize individual protons and so the picture emerging from spectroscopic analyses has been ambiguous, with some data supporting a Fe(IV)=O formulation and other data supporting Fe(IV)–OH. X-ray crystallographic work later followed. Long Fe–O bond lengths were reported and, as an indirect measure of protonation state, were interpreted as consistent with Fe(IV)–OH. But photoreduction is now known

to affect X-ray structures, and in the case of ferryl intermediates leads to reduction of the heme. Early structures were thus undermined. Photoreduction problems likely affect some of the spectroscopic experiments as well[11,12].

We have approached the question in a completely different way. Using neutron diffraction, hydrogen and deuterium atoms are directly visible, and photoreduction does not occur at all[13]. Thus, if the considerable difficulties of a neutron crystallographic experiment on a reactive enzyme intermediate trapped at low temperature can be overcome, then the approach is potentially transformative. In this work, we have used the approach to examine the Compound II intermediate of a heme peroxidase.

## Results

**Comparison of relevant peroxidases.** Ascorbate peroxidase (APX) catalyses the $H_2O_2$-dependent oxidation of ascorbate using the Compound I and Compound II intermediates. APX has high-sequence identity to cytochrome c peroxidase (CcP), which has served as a benchmark for heme enzyme catalysis over many years. But the experimental difficulty of isolating Compound II is simplified considerably by working with APX because Compound I in APX exists as a ferryl heme and a porphyrin π-cation radical[14,15], and is distinct from its Compound II, which contains only a ferryl species. This is not the case in CcP, as Compound I of CcP contains a ferryl heme and a tryptophan radical[16], which is not easily differentiated from its Compound II (ferryl heme only) in ultraviolet-visible spectra.

**Stopped flow and EPR spectroscopy.** Reaction of ferric APX with m-chloroperbenzoic acid (m-CPBA) initially yields a Compound I intermediate ($\lambda_{max} = 409, 527, 575^{sh}, 649$ nm; Fig. 1a); this decays rapidly to a Compound II intermediate ($\lambda_{max} = 415, 528, 558$ nm; see Fig. 1a and Supplementary Fig. 1), which is stable over long (500 s) timescales. The spectra of both Compound I and Compound II are pH-independent (Supplementary Fig. 2a,b).

Electron paramagnetic resonance (EPR) spectra confirm these observations (Fig. 1b). Initial formation of Compound I as the first intermediate on reaction of ferric APX with m-CPBA over

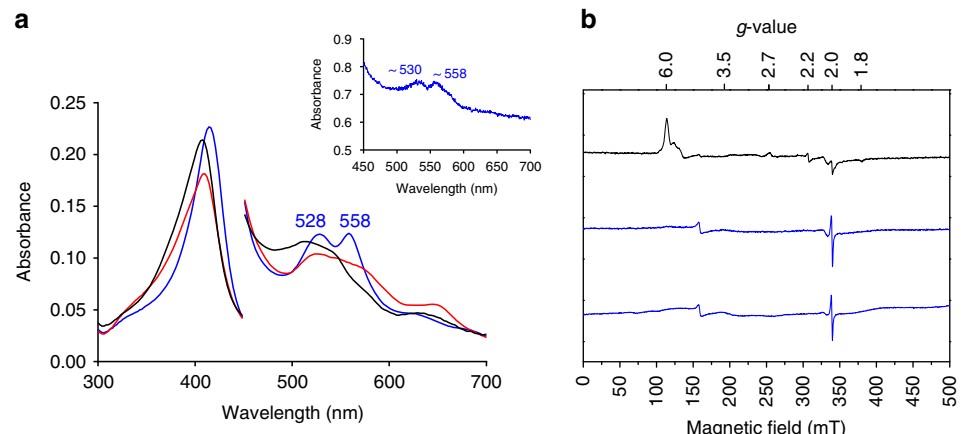

**Figure 1 | Formation of Compound II.** (**a**) Spectra obtained on reaction of ferric APX (black spectrum, at $t = 0$) with m-CPBA (10 equivalents), monitored over 100 s, showing formation first of Compound I (red spectrum) and then Compound II (blue spectrum). Absorbance values in the visible region have been multiplied by a factor of four. Conditions: 10 mM sodium phosphate, 150 mM KCl pH 7.0, 10.0 °C. Inset: Single crystal ultraviolet-visible spectra (100 K) of Compound II formed by reaction of ferric APX with m-CPBA showing the characteristic peaks ($\sim$530 and $\sim$560 nm) in the visible region. (**b**) 9 GHz EPR spectra of a solution of ferric APX (top spectrum, black), with the expected high-spin ($g_\perp = 6$ and $g_\parallel = 2$) and low-spin ($g_1 = 2.69$, $g_2 = 2.22$, $g_3 = 1.79$) heme resonances indicated; The spectrum of Compound II (middle spectrum, blue) prepared by reaction of ferric APX with 20 equivalents of m-CPBA and flash-frozen after 40 s; The bottom spectrum is same sample as the middle spectrum but recorded after 20 days. The spectra in Supplementary Figs 1 and 3b show that the pattern of reactivity with $H_2O_2$ is the same as with m-CPBA, giving the same Compound II species, in solution and in single crystals. Spectra were recorded at 7.5 K, 0.4 mT modulation amplitude, 1 mW power, 4 scans, 2,048 points.

short (7 s) timescales is apparent (new axial resonance at $g = 3.54$; Supplementary Fig. 3a). Over longer timescales (40 s), the ferric signals disappear and no new resonances are observed in Compound II (except for a very minor component from a radical signal ($<5\%$)) (Fig. 1b). Our spectra are in agreement with early EPR analyses on APX-I and APX-II (ref. 14). There is no evidence for a low-spin, paramagnetic Fe(III)–OH species in the EPR spectrum of Compound II (see also Supplementary Fig. 1g,h). These spectra demonstrate that the Compound II species is EPR-silent, confirming the presence of a ferryl heme. The spectrum of Compound II is stable when stored at 77 K over 20 days (Fig. 1b).

**Single crystal analyses**. Having characterized the reactivity in solution (as above), we then produced Compound II *in crystallo* and examined the crystals using single crystal microspectrophotometry and X-ray fluorescence. Single crystal microspectrophotometry at 100 K on crystals of ferric APX reacted with *m*-CPBA showed absorption peaks for Compound II in the visible region ($\lambda_{max} = 530$, 560 nm; Fig. 1a, inset) that agree with those in solution (Fig. 1a). The data indicate a high-percentage conversion in the crystal, and show that the predominant species is Compound II; we cannot rule out the presence of minor amounts of unreacted ferric enzyme (which are likely to be ferric hydroxide at this temperature). To confirm the oxidation state of the iron in the crystal, Fe K-edge X-ray fluorescence spectra were collected on a single crystal of Compound II. This showed a shift of the edge to higher energy compared with crystals of the ferric enzyme (Supplementary Fig. 3c), consistent with an increase in the iron oxidation state (from III to IV). Together, these spectra confirm the formation of Compound II and the presence of Fe(IV) in the crystal.

**Neutron diffraction**. The neutron structure of Compound II, prepared as above by reaction with *m*-CPBA, was solved at 100 K using data to 2.2 Å resolution. Data and joint X-ray/neutron refinement statistics are shown in Supplementary Table 1. Neutron and X-ray maps in the region of the heme are shown in Fig. 2a,b, respectively. Detailed views of the overall protein structure, individual active site residues and hydrogen bonding structure in the active site are shown in Supplementary Fig. 4. On the distal side of the heme, Trp41 is deuterated on Nε, and Arg38 is fully deuterated and is orientated away from the ferryl heme

(previously referred to as the 'out' position[17]). The distal histidine residue, His42, is doubly protonated (on Nε and Nδ).

An $F_o$–$F_c$ neutron map calculated by omitting the ligand on the distal side of the heme iron shows a positive difference density peak that was interpreted as arising from OD (Fig. 2c). The identity of this ligand was supported by calculations using an oxygen atom alone (that is, as in Compound I (ref.18)), which resulted in a positive difference peak at the D position (Supplementary Fig. 5a). There is no evidence from the maps for water occupancy at the distal site; if the distal site was modelled as $D_2O$, the other D site showed negative difference density (Supplementary Fig. 5b). No residual difference density was observed after refinement as OD.

**Multi-crystal X-ray crystallography**. The measured Fe–O bond length is 1.88 Å (Fig. 2c). This is longer than would be expected for an unprotonated Fe(IV)=O double bond species[19] and this distance is in the range expected for a single bonded Fe(IV)–OH species. It has been previously established using multi-crystal X-ray methods (to minimize photoreduction), that the corresponding bond length in C*c*P-I is 1.63 Å[17]. In this paper, independent verification of the Fe–O bond length in Compound II using the same multi-crystal X-ray methods (see 'Methods' section) yielded a bond length of 1.87 Å (Supplementary Fig. 5c), which agrees with the neutron data above (Fig. 2c) and with previous X-ray data for APX-II (ref. 17). Putting aside for one moment the caveats that bond lengths in X-ray experiments are not determined precisely at this resolution, nor indeed that they report directly on protonation state, the salient point is that the Fe–O bond length measured here for Compound II is longer than that of Compound I of C*c*P determined by reliable X-ray determinations (that is, where photoreduction has been minimized)[17,20]. There is also close agreement with recent Fe–O bond length determinations in Compound II of P450 (1.84 Å (ref. 21)).

## Discussion

Taken together, the data from the crystallographic and spectroscopic experiments are all consistent with the presence of an Fe(IV)–OH species in Compound II of APX. A broader comparison with the Compound I and Compound II intermediates of the related P450 enzymes is relevant (Fig. 3). The two classes of heme enzymes differ in their heme axial ligation—the P450s contain a Cys thiolate ligand on the proximal side of the

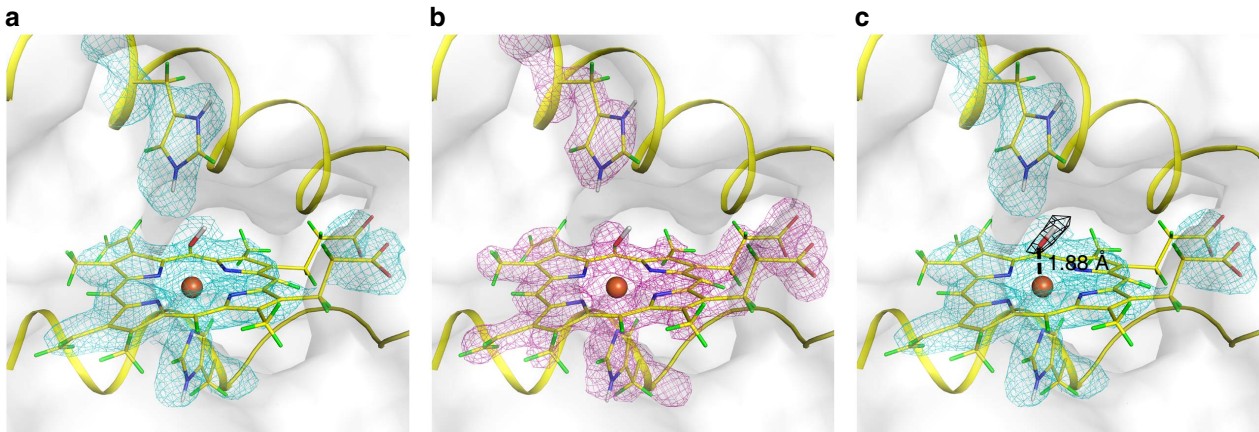

**Figure 2 | Neutron crystal structure of Compound II.** (**a**) Nuclear scattering density is shown in cyan (contoured at 1.5 σ). (**b**) Electron density is shown in magenta (contoured at 1.5 σ). (**c**) The neutron $F_o$-$F_c$ difference density calculated by omitting the distal ligand is shown in black (contoured at 3.0 σ), this is also shown as a stereo image in Supplementary Fig. 6. The O atom of the OD is positioned at 1.88 Å from the heme iron. Colour scheme: hydrogen—green; deuterium—white; carbon—yellow; oxygen—red; nitrogen—blue; and iron—brown sphere.

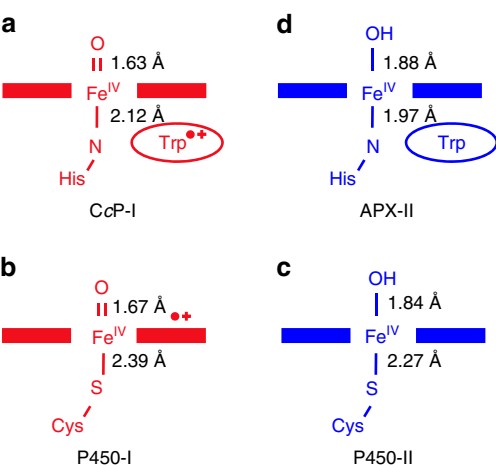

**Figure 3 | Comparison of Compound I and Compound II intermediates in peroxidases and P450s.** (**a**,**b**) Compounds I (shown in red) in CcP (ref. 18) and in both P450 and APO (refs 22,23) are Fe(V)=O species. (**c**) In P450 and APO, there is a large increase in p$K_a$ of the ferryl group going from Compound I to Compound II (the latter shown in blue), and the ferryl group is a protonated Fe(IV)–OH (refs 21,24). (**d**) Compound II of APX is an Fe(IV)–OH species, as presented in this paper. Bond lengths for CcP-I are from ref. 17 and for P450-I and P450-II are from refs 21,26.

heme, while the peroxidases have a histidine ligand instead. Compound I of cytochrome P450 has a radical on the porphyrin or the thiolate ligand (or delocalized between the two), and the ferryl heme is unprotonated[22]. Compound I in two closely related thiolate-ligated enzymes—aromatic peroxygenase (APO) and chloroperoxidase—are also assigned as Fe(IV)=O/porphyrin π-cation radical intermediates[7,23]. There is no information yet on a histidine-ligated peroxidase with a Compound I carrying a porphyrin π-cation radical. The best information comes from the atypical CcP (containing a Trp radical instead[16]), which also contains Fe(IV)=O (ref. 18).

Compound II in the bacterial CYP-158 P450 enzyme is a protonated Fe(IV)–OH species[21]. This pattern of unprotonated/protonated ferryl in Compound I/II in P450 is mirrored in both APO and chloroperoxidase[24,25] (Fig. 3). This means that there is a large 'uplift' in the ferryl p$K_a$ from Compound I (Fe(IV)=O) to Compound II (Fe(IV)–OH), which leads to the protonation event. What causes this shift in p$K_a$ of the ferryl heme in these thiolate-ligated systems is not clear, although bond length changes to the axial ligand[26], other hydrogen bond changes in the heme active site, and even distortions of the heme may all play a role as part of a complex mixture of variables that, together, dictate the nature and reactivity of the ferryl heme.

It has been widely assumed that formation of a Fe(IV)–OH species is not possible in the peroxidases, but our data demonstrate that it is. This offers a different perspective on the utilization of these ferryl intermediates across the family of heme enzymes. It remains to be seen whether other members of the peroxidase family behave similarly, but there is recent evidence that the ferryl heme could be protonated under certain conditions in other histidine-ligated heme enzymes[27,28]. The mechanisms of proton delivery are poorly understood. They are likely to involve networks of hydrogen-bonded residues and water molecules that connect the distal iron ligands all the way to the solvent surface[28–32]. As further new information emerges, the data presented here will help to lay a foundation for understanding the subtle and varied mechanisms that control proton delivery and biological reactivity in cytochrome P450 and other catalytic heme enzymes.

## Methods

**Protein expression and purification.** Soybean cytosolic APX, in a pLEICS-03 vector carrying kanamycin resistance and a TEV cleavable N-terminal His tag, was expressed in *E. coli* BL21 (DE3) cells. Cells were grown in 2-YT media for 16 h at 37 °C and protein expressed without induction with IPTG. Cells were collected by centrifugation (4,000*g* at 277 K for 20 min) and protein purified as previously described[33–35].

**Stopped-flow kinetics.** Pre-steady state stopped-flow experiments were carried out using an Applied Photophysics SX.18MV stopped-flow spectrometer. All experiments were carried out at 10 °C unless otherwise stated, using 10 mM sodium phosphate buffer, 150 mM KCl pH 7.0 (for APX) or pH 6.5 (for CcP). Spectral deconvolution was performed by global analysis and numerical integration methods using Pro-Kineticist software (Applied Photophysics Ltd). In all kinetic experiments, Compound II was formed in 93–95% yield.

Formation of Compounds I and II was followed in single mixing mode by mixing ferric enzyme (typically 2–3 μM) with 2–20 equivalents of $H_2O_2$ or *m*-chloroperbenzoic acid (*m*-CPBA) and time-dependent spectral changes monitored using a photodiode array detector. Data were fitted to a one-step model A→B, where A = ferric enzyme and B = Compound I (for Compound I formation) and A = Compound I and B = Compound II (for Compound II formation).

Compound II was also generated directly, under anaerobic conditions, by reaction of peroxide with the ferrous form of the enzyme. For these experiments, the sample handling unit of the stopped-flow instrument was housed in an anaerobic glove box (Belle Technology Ltd., [$O_2$] < 5 p.p.m.) and was used in the single mix mode. Ferrous enzyme, produced by titration of ferric APX (5–10 μM) with 2–5 equivalents of sodium dithionite, was mixed with 10 equivalents of $H_2O_2$. Time-dependent spectral changes accompanying Compound II formation were followed and data analysis carried out as outlined above.

To avoid enzyme instability problems (below pH 4.5 and above pH 11.5), the pH-jump method was used to investigate the pH-dependence of the spectra of Compounds I and II. Enzyme samples were prepared in water, adjusted to pH 7 with trace amounts of phosphate buffer (5 mM, pH 8.0). For Compound I formation, the enzyme was mixed with a stoichiometric amount of $H_2O_2$, which was made up in a buffer of twice the desired final concentration. The buffers used were citrate–phosphate in the pH range 4.0–6.0 (0.2 M), sodium carbonate–bicarbonate buffer in the range 8.0–10.5 (0.2 M) and sodium hydrogen phosphate (0.2 M) at pH 11.5. For Compound II formation, the sequential mix method was used and the enzyme was first mixed with $H_2O_2$ (also prepared in water at pH 7.0 as outlined above), the reaction allowed to age for 80 s to enable complete conversion to Compound II before a second mix with 0.2 M buffer. In all experiments, the pH of the solution was measured after mixing to ensure consistency.

**EPR spectroscopy.** Continuous-wave EPR spectra were recorded at 9.4 GHz on a Bruker EMX spectrometer with a Super-high-Q rectangular cavity and an Oxford ESR-900 liquid helium cryostat. The operating conditions are stated in the Figure legends. Samples of APX in solution were prepared in 10 mM potassium phosphate buffer, 150 mM KCl, pH 7.0. Compounds I and II were prepared by manually mixing ferric APX (370 μM) with an equivalent volume of *m*-CPBA or $H_2O_2$ solution directly in 4 mm quartz EPR tubes, followed by flash freezing in liquid nitrogen. We found that APX crystals are not large enough for single crystal EPR experiments because the absolute intensity of the EPR resonance for APX is too low.

**Crystallization.** Crystallization of APX was a modification of previous procedures and gave larger crystals (some as large as 1 × 0.6 × 0.4 mm$^3$) than previously (typically 0.15 × 0.075 × 0.075 mm$^3$ (refs 31,36). Crystals were grown by vapour diffusion hanging drops made up of 2–4 μl protein (20 mg ml$^{-1}$ in 10 mM potassium phosphate pH 7.0, 150 mM KCl) and an equal volume of precipitant (2.25 M $Li_2SO_4$, 0.1 M HEPES pH 8.3–8.9). The drop was allowed to equilibrate with 700 μl of precipitant. The crystals appeared in 2–14 days. Deuteration of APX was carried out by crystallizing the protein with the mother liquor made up with $D_2O$. Once the crystals were fully grown, the crystals were transferred and kept in the mother liquor with $D_2O$ until needed. Formation of Compound II in APX crystals for X-ray and neutron data collection was achieved by soaking the crystals in *m*-chloroperbenzoic acid (*m*-CPBA, 0.2 mM) for ∼40 s at 4 °C and then cryo-cooling crystals in liquid nitrogen.

**X-ray data collection.** The X-ray structure of Compound II of APX was solved by merging the first 10° of data from 10 different crystals. The data sets were collected in-house at 100 K using CuKα radiation (λ = 1.5418 Å) from a Rigaku MicroMax 007HF generator. Crystals of APX were reacted with *m*-CPBA as above and 20 images of 0.5° oscillation with 1 s exposure per image were recorded on a Rigaku Saturn 944+ detector to a resolution of 1.8 Å from each crystal. Data were indexed using iMOSFLM[37] then scaled and merged using AIMLESS as part of the CCP4 suite[38]. The statistics for the X-ray data collection are shown in Supplementary Table 1. X-ray fluorescence spectra on single crystals of ferric APX and Compound II were collected using the Vortex detector at Beamline I04, Diamond Light Source.

**Neutron data collection.** Preliminary experiments at the BIODIFF beamline[39] at the FRM II research reactor in Munich were used to test the experimental protocols, while full data collection took place using the LADI-III beamline[40] at the Institut Laue-Langevin (ILL), Grenoble. At the wavelengths/energies ($\geq 1$ Å, $\leq 81$ meV) typically used for crystallographic experiments, neutrons do not cause any observable radiation damage effects (discussed in ref. 13). For data collection, here we used 'cold' neutrons with $\lambda$ of 3.2–4.2 Å, corresponding to energies in the range 8–4.6 meV. A large ($0.7 \times 0.5 \times 0.4$ mm$^3$) single crystal of APX was reacted with m-CPBA as above to form Compound II and was then directly cryo-cooled to 100 K in the N$_2$ gas cryo-stream of the LADI-III instrument. Quasi-Laue neutron diffraction data extending to 2.2 Å resolution were collected at 100 K. As is typical for a Laue experiment, the crystal was held stationary at a different $\phi$ setting for each exposure. In total, 15 images were collected (with an average exposure time of 22.3 h per image) from two different crystal orientations. The neutron data were processed using the program LAUEGEN modified to account for the cylindrical geometry of the detector[41]. The program LSCALE[42] was used to determine the wavelength-normalization curve using the intensities of symmetry-equivalent reflections measured at different wavelengths. No explicit absorption corrections were applied. These data were then merged in SCALA[38]. The statistics for the neutron data collection are shown in Supplementary Table 1 in the supporting information.

**Structure refinement.** The crystals used for the data collections were iso-morphous with the previously published Compound II X-ray structure (PDB ID: 2XIF). All solvent molecules and the ligand were removed from 2XIF, which was then used as the starting model. The X-ray structure was solved and refined first, then the joint X-ray and neutron refinement was carried out with PHENIX[43,44]. H- and D-atoms were added with the program ReadySet[44] and D$_2$O molecules were added based on the neutron F$_o$–F$_c$ map. The model building was completed with Crystallographic Object-Oriented Toolkit software[45]. Determination of the identity and the position of heme ligand were based on the neutron data only. Joint X-ray/neutron structural refinement statistics are given in Supplementary Table 1.

**Single crystal spectrophotometry.** Absorption spectra of single crystals of Compound II of APX, obtained as outlined above, were collected at 100 K using an Ocean Optics Maya 2,000 PRO spectrometer, with an Ocean Optics DH-2000-BAL UV-VIS-NIR light source and a Humamatsu S10420 FFT-CCD back thinned detector with fibre optic coupled to 80 mm diameter 4X reflective lenses (Optique Peter, Lentilly, France) and mounted with a custom mount on Rigaku Raxis IV $\phi$ drive. The temperature was maintained at 100 K with an Oxford Cryosystems cryostream. Absorption spectra were acquired by means of the Ocean Optics SpectraSuite software. Single crystal microspectrophotometry was carried out as previously described[18] on single crystals of D$_2$O-exchanged ferric APX that had been reacted with m-CPBA (0.2 mM) or H$_2$O$_2$ (0.2 mM) for 40 s at 4 °C as above, followed by cryo-cooling at 100 K.

**Data availability.** The authors declare that the data supporting the findings of this study are available within the article and its Supplementary Information. Atomic coordinates and diffraction data have been deposited in the Protein Data Bank (accession codes 5JPR and 5JQR).

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

## Acknowledgements

We benefitted from discussions with D. Leys, D. Goodin and J. Groves. This work was supported by BBSRC (grant BB/K015656/1 to PM/ER), The Wellcome Trust (grant WT094104MA to PM/ER), the EPSRC National EPR Facility and Service, an Institut Laue-Langevin (ILL) studentship (to CC). Beam time was awarded from the LADI-III beamline, ILL and the BIODIFF beamline at FRM II. We acknowledge a Diamond award (MX 103690) to the UK Midlands BAG for I04 time.

## Author contributions

P.C.E.M., M.P.B., A.J.F and E.L.R. designed the research; H.K., J.B., C.M.C., A.J.F., T.E.S., A.O., M.P.B., J.M.D., P.A. and P.C.E.M. performed research; all authors analyzed data; E.L.R., M.P.B. and P.C.E.M. wrote the paper, with contributions from all authors.

## Additional information

**Competing financial interests:** The authors declare no competing financial interests.

**Publisher's note**: 

