## [Peer Review File · Nature Communications]

Reviewers' comments:

Reviewer #1 (Remarks to the Author):

This is an important milestone paper on the topic of the mechanism of action of heme based oxidases. Visualizing the hydrogen (or deuterium) atoms on intermediates is a great use of neutron crystallography in this case. The evidence presented is compelling. It is not so clear about the electronic state of the X-ray structure, where redox changes are very commonly seen in heme-iron complexes. The authors have tried to be careful with dose and radiation damage control by using fresh crystals but having spectroscopy on the crystals is really needed to nail this down. Nevertheless, the paper can stand on the neutron work. I find the paper to be well written and illustrated, but the key densities around the distal oxygen atom are still a bit hard to see without zooming in. Perhaps the figure could show less of the environment and more of the distal ligand since this is the main point of the paper.

Reviewer #2 (Remarks to the Author):

Unquestionably this work addresses an important issue in the P450 field, i.e. the precise nature of the key intermediate in catalysis, including its protonation state. This work employs neutron diffraction (on which I am not an expert but am aware of), in part to avoid the problem of photo reduction in X-ray analyses.

The coupling of spectroscopic and neutron results as single crystals is crucial.

With respect to the generation of Compound II in the crystals with M-CPBA, can time course data be given please. Were other oxidising agents than peroxide tested?

What is known about neutron damage? Is there a possibility that it could result in artefacts?

I think the calculations support rather than verify the experimental data.

As a non-neutron expert I would like to see a clear description of the direct evidence for OD, what contributes to the 'interpretation' of the density in fig. 2C. Could it be a mixture of $Fe=O$ and $Fe-OH_2$?

Non-experts would appreciate a brief description of how protonation occurs and its relevance within the context of P450 catalysis.

A brief description of the neutron results with respect to the porphyrin metal would be useful - does the position of the metal change? Are the crystals competent with respect to further reaction? This is a key question as to whether a true intermediate has been characterised (as implied in the abstract).

Overall from an enzymology perspective this is potentially a very interesting study. As indicated above I think it needs reviewing by a neutron expert, but I would be very much in favour of publication, if the issues above can be addressed.

Reviewer #3 (Remarks to the Author):

The manuscript by Kwon et al. reports on the nature of the ferryl species that is a common component of heme-containing enzymes. More precisely, the question to be answered is whether the ferryl is a $Fe(IV)=O$ or a $Fe(IV)-OH$ species. Previous structural and spectroscopic work has not provided a satisfactory answer to this question. The authors have used neutron diffraction to determine the structure of the ferryl intermediate in a heme peroxidase to 2.2 Å resolution. Using this structure, along with kinetic, EPR and single-crystal fluorescence analyses, Kwon et al. conclude that the reactive intermediate in these enzyme is a $Fe(IV)-OH$ species.

This is an important result that underlines the fact that in many enzymatic reactions the most

effective way of determining the structure of a catalytic intermediate is its direct observation in a crystal.

The manuscript could be significantly improved if:

- A scheme of a typical heme iron enzyme catalytic cycle is included (maybe as supplementary data). Many interested readers may not be very familiar with the reaction(s).

- The structure of a representative heme iron enzyme is depicted. Figures 2, S4 and S5 show the heme and some surrounding atoms but it would be nice to be able to place the prosthetic group within the protein (especially when the "distal heme pocket" is referred to (legend to Fig. S4)).

Other comments:

- "produced Compound II in crystallo and examined the crystals..." Wouldn't it be simpler to say: "grew crystals of Compound II and examined them..."

- "APX crystals are not large enough for single crystal EPR experiments". This is a surprising assertion given the fact that crystals large enough for neutron diffraction ($1.0 \times 0.6 \times 0.4$ mm³) were available. By the way it is mm³ and not mm.

Reviewer #4 (Remarks to the Author):

Moody, Raven and coworkers investigate the mechanism as it relates to the oxidation state of a ligand bound heme peroxidase using a Spectra of spectroscopic methods and neutron diffraction. Previous published data using x-ray crystallography has provided ambiguous results for the bleaching caused by radiation damage. In the current study, a carefully thought out procedure has been developed to trap an intermediate at hundred degrees Kelvin so that the intermediate is well populated as a major species. Although the full power of neutron diffraction is when experiments are conducted at room temperature since the method does not cause any radiation damage, here freezing crystals at hundred degrees Kelvin is required to maintain the intermediate. The nuclear density shows a difference positive peak that has been model as OD suggesting that the porphyrin ring contains Fe⁴-O intermediate.

The authors have modeled the DOD molecule to show that the extra D lies in negative density suggesting that indeed it is an OD atom. An early paper by Niimura and coworkers published in Protein Science on an atomic resolution insulin neutron diffraction structure have shown that there are disordered DOD molecules that had density in the shape of a cigar rather than the typical triangle or trapezoid. Hence, this is not a completely convincing argument. The authors should further consider this possibility. The resolution of this structure is reported at 2.2 Angstrom while the highest resolution shell only contains data that is 48% complete. Usually one has to have 50% complete this in order to claim such a resolution. The authors should change the resolution where there is 50% data present but can state the 2.2 Angstrom data was observed. Moreover, the neutron R_{free} is 31%.. Is this a little too high?

The manuscript is well written and once my concerns are addressed I will be happy to review the next revision.

Minor concerns

The supplementary figures do not appear in order

Reviewer 1.

1. *This is an important milestone paper on the topic of the mechanism of action of heme based oxidases. Visualizing the hydrogen (or deuterium) atoms on intermediates is a great use of neutron crystallography in this case. The evidence presented is compelling.*

Thank you for the comment.

2. *It is not so clear about the electronic state of the X-ray structure, where redox changes are very commonly seen in heme-iron complexes. The authors have tried to be careful with dose and radiation damage control by using fresh crystals but having spectroscopy on the crystals is really needed to nail this down. Nevertheless, the paper can stand on the neutron work.*

Neutron diffraction is non-ionising so does not lead to redox changes and we provided a large amount of experiments and spectroscopy (including on the crystals) to demonstrate the oxidation state of the iron. See for example Figures 1a, b (including the inset, showing the single crystal spectra), Figs. S1-S3 (including the X-ray fluorescence experiments, S3, on the crystals).

3. *I find the paper to be well written and illustrated, but the key densities around the distal oxygen atom are still a bit hard to see without zooming in. Perhaps the figure could show less of the environment and more of the distal ligand since this is the main point of the paper.*

We have re-examined the figures. There are two issues – colour and scale.

Colour. The most important figure (Figure 2C, in black) is very clear. We appreciate that Figure 2A might not be as clear, perhaps also Figure 2B, but these are less important. The combination of colours becomes limiting when presenting multiple sets of neutron and X-ray data. Looking back at the history of the manuscript, we tested 22 versions of the figures before settling on the colours presented here. No combination of colours / orientation is perfect – we picked what we considered to be the best.

Scale. In terms of focusing on the distal ligand only - heme proteins are always presented in the context of the “heme pocket” and we prefer to stick to this convention (which the community will recognize), otherwise the iron looks a bit homeless. During preparation of the figures we already deleted two fairly recognisable proximal residues (Trp179 and Asp208) from the “bottom” of the picture to try to focus just on the distal region. After reading the reviewer’s comments, we also prepared new versions of these figures showing just the iron, but in these images we also need to chop parts of the heme from the picture (which looks odd) and there is no “context” to the iron location in terms of the rest of the pocket. Note that reviewer 3 (reviewer 3, point 3, below) asks for more of the protein to be shown, not less.

So on balance we decided to change S4 (below, to address reviewer 3) and keep Figure 2 as it is. We hope that this is acceptable for reviewer 1.

Reviewer 2.

1. *Unquestionably this work addresses an important issue in the P450 field, i.e. the precise nature of the key intermediate in catalysis, including its protonation state. This work employs neutron diffraction (on which I am not an expert but am aware of), in part to avoid the problem of photo reduction in X-ray analyses.*

Thank you for the comment.

2. *The coupling of spectroscopic and neutron results as single crystals is crucial.*

Note our response to reviewer 1 above.

3. *With respect to the generation of Compound II in the crystals with *m*-CPBA, can time course data be given please.*

Time courses in crystals are not easy to generate because the reaction with *m*-CPBA is very fast. We showed time courses in solution in Fig. S1. In crystals, we did try reactions with *m*-CPBA over different reaction times before settling on the conditions reported in the methods, but in all cases we see immediate formation of CII in the crystal. Note that we have now also added time courses of crystals redissolved in solution to Figure S1g (point 13, below).

4. *Were other oxidising agents than peroxide tested?*

Some groups use just *m*-CPBA in these reactions (for example in the P450 papers). We used both *m*-CPBA and peroxide in our work.

The only other oxidizing agents that we might have tried would have been modified peroxides (e.g. t-Bu-OOH) but we know that these give the same CII species so there wasn't any need for us to explore this option.

5. What is known about neutron damage? Is there a possibility that it could result in artefacts?

We did have an explanation of this in our previous paper (ref 17) and we did not wish to labour that point again in the current paper so we had one line in the Introduction which said "Using neutron diffraction, hydrogen and deuterium atoms are *directly* visible, and photoreduction does not occur at all".

To re-state, neutrons are non-ionizing, and at the wavelengths/energies ($\geq 1 \text{ \AA}$, $\leq 81 \text{ meV}$) typically used for crystallographic experiments they do not cause any observable radiation damage effects. For data collection here we used neutrons with λ of 3.2 – 4.2 \AA , corresponding to energies in the range 8 – 4.6 meV. We have added this more detailed comment to the Methods.

6. I think the calculations support rather than verify the experimental data.

Agreed. We have changed "was verified" to "was supported" in the text on p3 (see also below).

7. As a non-neutron expert I would like to see a clear description of the direct evidence for OD.....

We worried about this when writing the manuscript. We have consulted with the neutron experts at ILL, and we are advised that the original statement was fairly expansive. We said: 'An $F_o - F_c$ neutron map calculated by omitting the ligand on the distal side of the heme iron shows a positive difference density peak that was interpreted as arising from OD (Fig. 2c). The identity of this ligand was verified (*now changed to "was supported"*) by calculations using an oxygen atom alone (*i.e.* as in Compound I¹⁷), which resulted in a positive difference peak at the D position (Supplementary Fig. 5a). There is no evidence from the maps for water occupancy at the distal site; if the distal site was modelled as D₂O the other D site showed negative difference density (Supplementary Fig. 5b). No residual difference density was observed after refinement as OD.'

We don't think we can add much to this statement that will make the description any clearer.

8. what contributes to the 'interpretation' of the density in fig. 2C.

We have addressed this above.

9. Could it be a mixture of fe=0 and fe-OH2?

We did consider this (at great length!). We have addressed this in the reply to reviewer 4, point 1, below. In addition, there is no evidence for water-bound heme (which is Fe(III) not Fe(IV)) in any of the spectroscopic data (which can easily differentiate mixtures of species).

10. Non-experts would appreciate a brief description of how protonation occurs

We agree that this would be helpful. The problem is that the mechanisms of proton delivery are *very* vague in the peroxidase and (to a lesser extent) the P450 literature. We did have a section on proton delivery in early versions of the paper, but we were unhappy with it as it seemed too speculative and we could not say anything concrete from our data that would help to move the field forward. We took external advice (informal peer review) prior to submission on how to handle this, and as a result of the feedback we decided to avoid opening a discussion on the subject of protonation events and removed that section. We felt that it was better to simply present the data as we observe it and then let the community mull over the results; our external reviewers agreed with us on this. We realize that this means that we don't talk specifically about *how* protonation occurs, but we did present Figure S4b (now S4c) which would give an indication of possible proton relays for anyone who was interested.

In response, we have re-introduced a new sentence at the end of the discussion on this. We hope the reviewer can appreciate the balance that we tried to strike here, and that our compromise situation is acceptable. We have tried to keep the paper focused on the observation and what the data report, and have kept to a minimum any speculation beyond that.

11. ... and its relevance within the context of P450 catalysis.

We believe that was already covered in our statement in the introduction: "This is important because the bonding interactions and the protonation state of the ligand bound to the iron controls the reactivity – and hence the biological usefulness – of each intermediate."

12. A brief description of the neutron results with respect to the porphyrin metal would be useful - does the position of the metal change?

This is an important question and it was raised when we showed our early data to colleagues. In our earlier work (ref 16), where we reported the X-ray structure of APX-CII formed by reaction with H₂O₂, we noted that the ferryl iron moves out of the heme plane by 0.15 \AA toward the distal histidine (when compared with the ferric enzyme), and the proximal histidine (His163) also shifts in the same direction. We do observe a similar but slightly smaller ($\sim 0.1 \text{ \AA}$) movement in this structure with respect to the ferric enzyme, but the resolution of this structure does not allow us to verify or describe it with confidence. Hence, we do not think that our data can shed light on this question.

13. Are the crystals competent with respect to further reaction? This is a key question as to whether a true intermediate has been characterised (as implied in the abstract).

The issue here is whether CII reverts back to a ferric species. We had presented the extensive data in Figure 1 (which was a summary of many control experiments in crystals and in solution) to demonstrate the

reliability/reproducibility of CII formation. It is not physically possible to test the crystals after the data collection in the neutron source, because of the difficulty of handling frozen crystals at the neutron facility. Instead, we had examined *m*-CPBA-soaked crystals that had been redissolved in buffer in solution (10 mM potassium phosphate pH 7.0, 150 mM KCl) – we found that these crystals give the same CII maxima as for CII in solution **and** that the CII species in the crystal reverts back to a ferric species in solution over the same timescales for CII in a solution experiment. (Remember that APX CII in the crystal has the same spectrum as APX CII in solution.) We didn't show those data because we were advised during our own peer review processes that Figure S1 was already getting quite detailed and that all the checks on the crystals that we carried out would be too specialised for all but a few readers of a general journal such as this one.

In response to the reviewer's comment. we have now added these data as an inset to Figure S1g, to now show the spectrum of a crystal of APX CII redissolved in buffer and the change in the spectrum of that species over time (showing conversion back to the ferric). We think this is a very convincing addition (thank you for the suggestion).

14. Overall from an enzymology perspective this is potentially a very interesting study. As indicated above I think it needs reviewing by a neutron expert, but I would be very much in favour of publication, if the issues above can be addressed.

Thank you for the comments.

Reviewer 3.

1. The manuscript by Kwon et al. reports on the nature of the ferryl species that is a common component of heme-containing enzymes. More precisely, the question to be answered is whether the ferryl is a Fe(IV)=O or a Fe(IV)-OH species. Previous structural and spectroscopic work has not provided a satisfactory answer to this question. The authors have used neutron diffraction to determine the structure of the ferryl intermediate in a heme peroxidase to 2.2 Å resolution. Using this structure, along with kinetic, EPR and single-crystal fluorescence analyses, Kwon et al. conclude that the reactive intermediate in these enzyme is a Fe(IV)-OH species. This is an important result that underlines the fact that in many enzymatic reactions the most effective way of determining the structure of a catalytic intermediate is its direct observation in a crystal.

Thank you for the comments.

2. A scheme of a typical heme iron enzyme catalytic cycle is included (maybe as supplementary data). Many interested readers may not be very familiar with the reaction(s).

The difficulty here is that it is not straightforward to show reaction mechanisms for the range of enzymes that we are discussing. While they all use the same intermediates (Compounds I and II) they have slightly different mechanisms. For a general audience, we felt that it was better to keep the message simple and to focus more plainly on the "common intermediates" which glues all the mechanisms together. The citations that we give at the beginning of the paper should direct the reader to the specialized explanations of individual mechanisms. We have adjusted the text at the start of the results to make it plain that APX used the same intermediates. We hope this is acceptable.

3. The structure of a representative heme iron enzyme is depicted. Figures 2, S4 and S5 show the heme and some surrounding atoms but it would be nice to be able to place the prosthetic group within the protein (especially when the "distal heme pocket" is referred to (legend to Fig. S4)).

This is reasonable, but it is the opposite of what reviewer 1 asked for (reviewer 1, point 3)! It wasn't clear whether reviewer 3 wanted this change in the main text or whether in S4. But bearing in mind reviewer 1's comments, we put the change into S4, so as not to distract from the "main point of the paper" which reviewer 3 commented on. Text and Figure have both been adjusted. Hopefully this compromise will satisfy both reviewers.

4. "produced Compound II in crystallo and examined the crystals..." Wouldn't it be simpler to say: "grew crystals of Compound II and examined them..."

Not really, because the crystals were reacted which is why we have said that we produced Compound II *in crystallo*.

5. "APX crystals are not large enough for single crystal EPR experiments". This is a surprising assertion given the fact that crystals large enough for neutron diffraction (1.0 x 0.6 x 0.4 mm³) were available.

Apologies, that was not clear. The absolute intensity of the EPR resonance for APX is much lower than that for CcP because APX has greater *g* anisotropy: CcP has a far more isotropic signal, and gives a relatively narrow Trp radical signal (high spin density) compared to APX, as Goodin and Poulos showed (our ref 13). The crystals of APX are also *much* smaller (by about ten times) than those for CcP on which we were able to obtain single crystal EPR spectra. This combination makes single crystal EPR unfeasible for APX. We have edited the text accordingly in the methods.

By the way it is mm³ and not mm.

This has been corrected in three places, thank you for spotting it.

Reviewer 4.

Moody, Raven and coworkers investigate the mechanism as it relates to the oxidation state of a ligand bound heme peroxidase using a Spectra of spectroscopic methods and neutron diffraction. Previous published data using x-ray crystallography has provided ambiguous results for the bleaching caused by radiation damage. In the current study, a carefully thought out procedure has been developed to trap an intermediate at hundred degrees Kelvin so that the

intermediate is well populated as a major species. Although the full power of neutron diffraction is when experiments are conducted at room temperature since the method does not cause any radiation damage, here freezing crystals at hundred degrees Kelvin is required to maintain the intermediate. The nuclear density shows a difference positive peak that has been model as OD suggesting that the porphyrin ring contains Fe4-O intermediate.

1. The authors have modeled the DOD molecule to show that the extra D lies in negative density suggesting that indeed it is an OD atom. An early paper by Niimura and coworkers published in Protein Science on an atomic resolution insulin neutron diffraction structure have shown that there are disordered DOD molecules that had density in the shape of a cigar rather than the typical triangle or trapezoid. Hence, this is not a completely convincing argument. The authors should further consider this possibility.

This is the same question as raised by reviewer 2 (point 9) above. To expand, the interpretation has to be considered in the context of the other experimental work presented in the paper. The electronic absorbance spectroscopy, EPR and X-ray fluorescence all support the formation of Compound II in the crystal; none of those spectra are consistent with a water molecule bound to the heme in the crystals and can only be assigned as Fe(IV)=O or Fe(IV)-OH (or D in this case).

The literature from Niimura and co-workers which analyses the observed shapes of D₂O density in neutron protein structures (Chatake et al., (2003) Proteins 50 516-523, Chatake et al., (2003) Acta Cryst D60 1364-1373) conclude that rotational disorder in waters is highly correlated with the B value of the oxygen atom. The B value of the O atom of the OD that we show (21.7 for the joint (X+N) refinement, 10.7 when refined with X-ray alone) does not correlate with a disordered water molecule.

2. The resolution of this structure is reported at 2.2 Angstrom while the highest resolution shell only contains data that is 48% complete. Usually one has to have 50% complete this in order to claim such a resolution. The authors should change the resolution where there is 50% data present but can state the 2.2 Angstrom data was observed. Moreover, the neutron R_{free} is 31%. Is this a little too high?

Quantifying the “resolution” of data is contentious and the definitions inconsistent and arbitrary, for example a cut-off at the point where the mean I/σ becomes lower than 3.0 (or 2.0) is widely used. To avoid this issue we have removed any “claim” to a specified resolution in the abstract. We believe the data statistics presented in supplementary Table 1 are much more informative and are sufficient. We have kept references in the document to “data used to 2.2 Å”.

It should be noted that the protocols used for the collection of neutron diffraction data often give lower data completeness than that seen for X-ray work. We also note that neutron structures typically have higher R factors than X-ray data and that there are a number of neutron structures in the PDB with R_{free} > 30%. Moreover, the majority of the protein structure the case of APX has already been well determined in our previous work. We are reluctant to arbitrarily exclude any useful data just to improve the statistics, preferring to use all the data we have available. We also wished to avoid the introduction of artefacts due to resolution truncation. In response to the reviewer’s comment, we have re-refined the structure using only data to 2.31Å (below, in which completeness is > 50%) and report the resultant statistics below. If the referee considers the original statistics we present are unacceptable, we will substitute these values and update the PDB entries accordingly.

Supplementary Table 1. Data collection and joint X-ray/neutron structural refinement statistics for APX-II (revised).

Space group	P4 ₂ 2 ₁ 2
Cell dimensions (Å)	82.095, 82.095, 75.162
Data collection (Neutron, quasi-Laue $\lambda = 3.2 - 4.2 \text{ \AA}$)	
Resolution	40.00 - 2.31 (2.43 - 2.31)
R _{merge}	0.18 (0.21)
I/σ	7.1 (4.7)
Completeness (%)	73.1 (50.0)
Redundancy	4.1 (2.4)
Data collection (X-ray $\lambda = 1.5418 \text{ \AA}$)	
Resolution	19.91 - 1.80 (1.84 - 1.81)
R _{merge}	0.16 (0.69)
I/σ	9.4 (2.1)
Completeness (%)	98.72 (92.0)
Redundancy	6.8 (4.0)
Refinement	
d _{min} (Neutron, Å)	2.31
d _{min} (X-ray, Å)	1.81
No. of reflections (Neutron)	8367
No. of reflections (X-ray)	23856
R _{work} /R _{free} (Neutron)	0.20/0.27
R _{work} /R _{free} (X-ray)	0.16/0.21
RMSD	
Bond length (Å)	0.01
Bond angle (°)	1.35

3. *The manuscript is well written and once my concerns are addressed I will be happy to review the next revision.*
We hope the reviewer is satisfied with the responses to this and the other concerns above.

4. *The supplementary figures do not appear in order.*

Noted. The reason why the supplementary figures might appear to be out of order is because the reference to Figure S1 and S3b is in the legend to Figure 1 (we did this so as not to disrupt the flow of text with too much detail in the main document). Because of this, the first mention of a supplementary figure in the main text is for Figure S2. In the legend, we have now directed the reader to Figure S1 – in this Figure S1, the reader is directed sequentially from S1a to figures S1b-f). Only a specialized audience will want this level of detail, and so we have simplified the legend to Figure 1 accordingly (see highlighted Figure 1 legend and S1 legend). We ask the office for clarification as to whether this is acceptable.

REVIEWERS' COMMENTS:

Reviewer #2 (Remarks to the Author):

The authors have given detailed and thoughtful responses to the issues raised by myself and the other reviewers. From an Fe-enzymology perspective, I think the work is a very widespread interest and I'm very happy to support publication. Congratulations to the authors on a very nice application of neutron diffraction.

Minor point – please add citation for the following statement o non ionising nature of neutrons in the collection section.

Reviewer #3 (Remarks to the Author):

Changes are OK.

Reviewer #4 (Remarks to the Author):

Raven and colleagues have responded to the reviewer's comments. With respect to my comment on the nuclear density not being able to unambiguously identify the OD atom they have argued that the evidence is in the multiplicity of the methods. If this is true they should clearly state this also mentioning that the nuclear density itself is not absolute evidence. With respect to my concerns of resolution limits they have recalculated a resolution of 2.33 Å in a new table which should be used in this publication as now there is data in the last resolution shell beyond 50% completeness. If these are implemented I will be happy to see this manuscript published.

Response to Referees

REFEREE COMMENTS

1. Minor point – please add citation for the following statement to non ionising nature of neutrons in the collection section.

We already had a reference to that in the text (ref 13) on p2. But to avoid confusion we have removed all reference to neutrons being non-ionising (in the methods) and we have changed the relevant sentence accordingly to “At the wavelengths/energies ($\geq 1 \text{ \AA}$, $\leq 81 \text{ meV}$) typically used for crystallographic experiments they do not cause any observable radiation damage effects (discussed in ¹³). For data collection here we used ‘cold’ neutrons with λ of $3.2 - 4.2 \text{ \AA}$, corresponding to energies in the range $8 - 4.6 \text{ meV}$.”

2. With respect to my comment on the nuclear density not being able to unambiguously identify the OD atom they have argued that the evidence is in the multiplicity of the methods. If this is true they should clearly state this also mentioning that the nuclear density itself is not absolute evidence.

We think that we had done this already in our statement “Taken together, the data are all consistent with the presence of an Fe(IV)-OH species in Compound II of APX”. But we have changed it to “Taken together, the data from the crystallographic and spectroscopic experiments are all consistent with the presence of an Fe(IV)-OH species in Compound II of APX.”

3. With respect to my concerns of resolution limits they have recalculated a resolution of 2.33 \AA in a new table which should be used in this publication as now there is data in the last resolution shell beyond 50% completeness. If these are implemented I will be happy to see this manuscript published.

We have replaced the table but note the resolution is 2.31 not 2.33 \AA .